# Properties and Factors of Cs_x_WO_3_ Slurry for Building Glass with High Visible Light Transmission and Outstanding Near-Infrared Insulation

**DOI:** 10.3390/ma17215196

**Published:** 2024-10-25

**Authors:** Yunpeng Liu, Yuqi Mu, Xihao Yang, Zhiyu Yao, Shaofeng Peng, Jincheng Shi, Wendi Tian, Yen Wei, Kangmin Niu

**Affiliations:** 1School of Materials Science and Engineering, University of Science and Technology Beijing, Beijing 100083, China; 15200096466@139.com (Y.L.); m202120432@xs.ustb.edu.cn (Y.M.); m202220506@xs.ustb.edu.cn (X.Y.); d202110270@xs.ustb.edu.cn (W.T.); 2Hebei Construction Group Corporation Limited, Baoding 071000, China; 13831298152@139.com (Z.Y.); 13313120223@189.com (S.P.); 15830897122@139.com (J.S.); 3Department of Chemistry, Tsinghua University, Beijing 100084, China

**Keywords:** tungsten bronze material, energy-saving glass, finite element method, laminated glass

## Abstract

This study is dedicated to the development of a new type of cesium tungsten bronze energy-saving laminated glass and explores its application in insulating glass combinations, offering innovative ideas and practical solutions for advancing energy-saving glass technology. Experimental results show that both Cs_x_WO_3_ (CWO) dispersions exhibit good visible light transmittance and near-infrared shielding properties, with CWO1 demonstrating superior shielding in the 650–950 nm range, attributed to differences in shape and size distribution and verified by simulations using the Drude–Lorentz model and the finite element method.

## 1. Introduction

With the continuous increase in building area, building energy consumption has gradually become a problem that must be paid attention to. According to statistics, the energy loss caused by glass accounts for 40% of the total loss of buildings. This is primarily because the traditional window glass allows most of the energy from sunlight to enter the room, which will increase the cooling pressure of buildings, resulting in substantial energy consumption [1,2]. According to the spectral data of solar energy distribution, 98% of energy is concentrated in the wavelength range of 200–2500 nm, of which the ultraviolet (UV) radiation in the 200–400 nm range accounts for approximately 5%, the visible light (Vis) in the 400–700 nm range for about 43%, and the near-infrared (NIR) light in the 700–2500 nm range for about 52% [3,4]. Therefore, in order to reduce energy consumption, scientists have researched and developed various energy-saving glasses for windows, including Low-E glass and nano-functional materials [5,6,7]. Low-E glass features a complex dielectric/metal/dielectric (D/M/D) coating structure [8], which reflects infrared light. However, the cost of precious metal raw materials and coating equipment is high; meanwhile, the oxidation resistance is poor, and the film structure is complex. In contrast, nano-functional materials are inorganic nanomaterials with special optical properties that are combined with polymer resin to make functional composites for application in glass. By utilizing the UV/NIR absorption effect of nano-functional materials, they can maintain clear vision and lighting conditions while reducing the energy of sunlight entering the room. This plays a role in heat insulation and energy savings, as illustrated in Figure 1. Common functional materials include indium/antimony doped tin oxide (ITO/ATO) and lanthanum hexaboride [9,10,11]. The ITO/ATO film has stable chemical properties, good adhesion with glass, and high visible light transmittance, but it cannot effectively shield NIR with a wavelength less than 1500 nm. The effectiveness of LaB_6_ in shielding NIR is also limited. Additionally, LaB_6_ is usually prepared from La_2_O_3_ and B_4_C in a reducing atmosphere of 1500 °C [12]. Consequently, both the production costs and the expense associated with the rare earth element lanthanum (La) are high.

Tungsten bronze represents a new class of materials that are ideal for energy-saving glass applications, and it is an important semiconductor material with the chemical formula M_x_WO_3_, where “M” represents variable ions such as hydrogen, ammonium, or alkali metal ions [13,14]. Due to its special crystallographic, electronic and optical properties, tungsten bronze has established a solid research foundation in various fields, including superconductors, optical devices, sensors, and catalysts [15,16,17,18]. In 2007, Takeda et al. [19] proposed that nanoscale tungsten bronze could be utilized as a solar light filtering material, which has propelled it to the forefront of research in areas such as optical selectivity and transparent thermal insulation. The fundamental structure of tungsten bronze is based on WO_3_, which forms a framework interconnected by octahedron and creates different types of gaps for M ions to occupy, thus forming the corresponding tungsten bronze materials. Tungsten bronze crystals predominantly exhibit cubic, tetragonal, and hexagonal structures [20], which are related to the type of doped ions, the synthesis method, and the synthesis temperature. The number of gaps in cubic and tetragonal tungsten bronze is large, and the size is small, which makes it easier to add more small ions such as H^+^, Li^+^, and Na^+^ [21]. Nevertheless, the hexagonal tungsten bronze has some large gaps, so it is easier to be doped with Rb^+^, Cs^+^, and NH_4_^+^. However, the doping amount is relatively low, with x usually being less than 1/3 [22]. In addition, there are also some co-doped tungsten bronze with different gaps doped with different cations, such as Na_x_Cs_y_WO_3_ [23,24]. The choice of doping ions and the amount of doping can significantly influence the optical properties of the material. The cesium tungsten bronze (Cs_x_WO_3_, CWO) with a hexagonal phase structure is the most in-depth researched material in the field of energy-saving glass.

A key characteristic of tungsten bronze is its selective absorption of UV and NIR light, which is attributed to three mechanisms: intrinsic absorption, local surface plasmon resonance (LSPR), and small polaron absorption. LSPR is the main absorption mechanism for NIR. In tungsten bronze, when metal cations (such as potassium or sodium) are incorporated into the structure of WO_3_, some free electrons are released. In this context, plasma in the crystal typically refers to the interaction between the free electrons and positive ions within the crystal. As the free electrons move through the crystal, they are influenced by the array of positive ions in the lattice, causing them to vibrate at a certain natural frequency. This frequency is an intrinsic property of the system and is usually determined by the electron density and dielectric constant. When the frequency of the incident light approaches this natural frequency, plasma resonance will occur [25]. At this point, incident light energy will be absorbed, thereby affecting the propagation of light. Specifically, the material has different dielectric constants for different light wavelengths. According to the Drude model, the LSPR absorption frequency and its influence on the dielectric constant are depicted in Equations (1) and (2) [26]:(1)$$\omega_{P}=\sqrt{\frac{Ne^{2}}{\varepsilon_{0}m_{e}}},$$
(2)$$\varepsilon_{D}(\omega )=\varepsilon_{\infty }(\omega )-\frac{\omega_{P}^{2}}{\omega^{2}-\omega \gamma },$$
where *ω_P_* is the LSPR frequency; *N* is the free electron concentration; e is the electronic charge; *m*_e_ is the electronic effective mass; *ε*_0_ and *ε*_∞_ are the permittivity of vacuum and high frequency, respectively; *ε_D_* is the relative dielectric constant; *γ* is the damping factor; and *ω* is the frequency of the incident light. Tungsten bronze materials contain a large number of free carriers and exhibit a significant LSPR effect. They primarily absorb NIR and convert it into heat [27]. For tungsten bronze crystals with an anisotropic structure, there are different *ω_P_* values along the directions of the cell’s a/b and c axes [28,29]. Kim et al. [26] found through experiments and simulations that the length-to-diameter ratio of hexagonal CWO nanorods has certain adjustability in LSPR absorption, similar to noble metal nanorods. Tracy et al. [25] discovered that the different media surrounding the particles can affect the absorption peak of CWO. It can be seen that the LSPR absorption effect is related to the shape, size, and medium of tungsten bronze particles.

Tungsten bronze materials are typically used at the nanometer scale to maintain transparency and to fully utilize the required properties. The dispersion effect of nanoscale tungsten bronze materials is closely related to the powder preparation process, dispersion process, and dispersant. Some scholars have conducted specific studies on these factors. For example, Liu et al. [30] ground and dispersed CWO prepared by a hydrothermal method and investigated the effects of the dispersion process, pH, and dispersant type on the particle size and properties of this kind of powder. However, there are still some deficiencies in the systematic study of the factors affecting the properties of CWO dispersion.

In this paper, the CWO powder was prepared using an improved solid-state method. Compared with commercially available powders, it was modified with various dispersants to improve the compatibility of CWO with different solvents, and the preparation of CWO nano dispersion in water and ethanol systems was realized. The factors affecting the dispersity and optical properties of the dispersion were discussed.

## 2. Materials and Methods

### 2.1. Materials

Cs_2_CO_3_, WO_3_, and W powders were provided by Sinopharm Chemical Reagent Co., Ltd. (Shanghai, China), as raw materials for CWO. The commercial CWO powder used for the comparative experiment was supplied by Jiangxi SunNano New Materials Technology Co., Ltd., Ganzhou, China. Several different dispersants were selected, as shown in Table 1. Dispersants 1 and 2 are anionic polyelectrolyte dispersants mainly composed of sodium polyacrylate and polyacrylate, respectively. Dispersant 7 is a cationic polyelectrolyte dispersant with secondary amine groups.

### 2.2. Methods

#### 2.2.1. Preparation of CWO

First, cesium carbonate (Cs_2_CO_3_), tungsten oxide (WO_3_), and tungsten powder (W) were thoroughly mixed and initially ground. The mixture was then ball-milled at a speed of 300 revolutions per minute (r/min) for 12 h. Subsequently, the resulting mixture was transferred to a tube furnace and annealed at 600 °C for 90 min under a nitrogen or nitrogen-hydrogen mixed atmosphere. After cooling, the desired CWO powder was obtained. The CWO powder prepared in this part was named CWO1. In addition, a commercially purchased CWO powder, designated as CWO2, was used for comparative experiments. The experimental process is shown in Figure 2.

#### 2.2.2. Preparation of Dispersion

An amount of 25 g of the dispersion medium was weighed out, and after incorporating a certain amount of dispersant, it was stirred until fully dissolved. Then, 1 g of CWO powder was added to a 30 mL zirconia ball mill jar containing grinding balls. Following this, the ball mill jar was placed into a planetary ball mill and ground at a speed of 600 r/min for a certain period of time to obtain the CWO dispersion.

#### 2.2.3. Characterizations

The powder was characterized using an X-ray diffraction instrument (Rigaku Ultima IV, Tokyo, Japan) to analyze its phase composition. The surface morphology and particle size were determined by scanning electron microscopy (SEM) (TESCAN MIRA LMS, Brno, Czech Republic). The surface functional groups of the particles were tested and analyzed using Fourier-transform infrared spectroscopy (Thermo Scientific Nicolet iS20, US-MA, Waltham, MA, USA). Thermal stability analysis was conducted using a thermogravimetric analysis and differential scanning calorimetry (TG-DSC) synchronous thermal analyzer (Mettler TGA/DSC1, Zurich, Switzerland), with a measurement temperature range of 30 to 600 °C, a heating rate of 10 °C/min, and N_2_ atmosphere.

The stability of the dispersion was evaluated using the sedimentation method. Prior to testing, the sample was ultrasonically dispersed for 15 min, then left to stand for 7 days under identical conditions, and the stability was evaluated by visually inspecting the degree of sedimentation. The particle size distribution of the powder and its dispersion was measured by a laser particle size analyzer (Malvern Zetasizer Nano ZS ZEN3600, Malvern, UK). The powder or dispersion was diluted to an appropriate concentration and then subjected to ultrasonic treatment for 5 min to ensure proper dispersion before testing. The transmission spectrum of the dispersion was tested with a spectrophotometer (Mapada UV-3000PC, Shanghai, China). The sample was diluted 40 times, ultrasonically dispersed for 5 min, and then introduced into the instrument for analysis.

#### 2.2.4. Model

The Wave Optics Module of the COMSOL Multiphysics 6.1 software, which employs the finite element method, was used to perform frequency-domain calculations on the Maxwell electromagnetic field equations for the wavelength range of 300–1100 nm. A harmonic background field was established along a specific direction, assuming that light of a particular wavelength irradiates the CWO particles. The shape of the particles was reconstructed according to the results of electron microscopy characterization, either as hexagonal prisms or spheres. Due to symmetry, a 1/4 model can be established.

The primary material parameters of the model are the refractive indices of the particles and the surrounding medium. The refractive index of the CWO particles is defined by the dielectric constant, which is closely related to the LSPR absorption phenomenon. This relationship is described using the Lorentz–Drude dispersion model, as shown in Equation (2).

For the complete hexagonal phase CWO crystal, these material parameter values exhibit anisotropy, as shown in Table 2, which are derived from the study by Hussain et al. [31]. The material parameters in the specific model depend on the shape of the particle growth. For hexagonal prism-shaped particles, the anisotropic parameters listed in the table are employed. For polycrystalline spherical particles, whose properties do not show significant anisotropy, the average of the axial and longitudinal parameters is taken as the isotropic model parameter.

## 3. Results and Discussion

### 3.1. Characterization of CWO Powder and Dispersion

#### 3.1.1. XRD

Firstly, the crystal structure and phase composition of the CWO were characterized using X-ray diffraction (XRD), as shown in Figure 3. It can be observed that all the diffraction peaks closely align with the hexagonal phase Cs_0.32_WO_3_ (PDF#83-1334). The peaks at 2θ values of 13.8°, 23.4°, 27.3°, 27.8°, 33.8°, 36.6°, 44.3°, 47.8°, 48.8°, 49.1°, 56.2°, 57.1°, 57.4°, 71.0°, and 78.8° correspond to the (100), (002), (102), (200), (112), (202), (212), (004), (302), (220), (204), (312), (400), (224), and (420) planes of hexagonal Cs_0.32_WO_3_, respectively. No significant extra peaks are observed, indicating a high purity of CWO.

In contrast, the narrower peak widths of CWO1 indicate higher crystallinity and larger primary crystal grain sizes. According to calculations using the Scherrer formula [32], the grain size of CWO1 exceeds 100 nm, which is beyond the applicable range of the formula. In comparison, the grain size of CWO2 is approximately 22 nm, indicating an extremely fine state. However, this calculation method cannot determine the secondary particle size resulting from particle agglomeration, thus further characterization of the powder’s particle size is required.

#### 3.1.2. XPS

The X-ray photoelectron spectroscopy (XPS) full spectrum analysis (Figure 4) shows that both powders contain the elements Cs, W, and O. Figure 4b presents the W4f spectrum of the powders, where two pairs of spin-orbit split peaks corresponding to W^6+^ and W^5+^ are observed. The area of the W^5+^ peak in CWO1 is relatively larger compared to that in powder 2. Figure 4c reveals that in addition to the fitting peaks corresponding to O-H and O-W chemical bonds, a noticeable third peak is present in powder 1. This suggests that CWO1 may contain a certain amount of oxygen vacancies, which can provide electrons, thereby increasing the amount of W^5+^. This is consistent with the phenomenon observed in Figure 4b [33]. This phenomenon facilitates the promotion of small polaron absorption.

#### 3.1.3. FT-IR

Figure 5 shows the Fourier-transform infrared spectroscopy (FTIR) spectra of the obtained samples within the wavenumber range of 4000–400 cm^−1^. As seen in the figure, both CWO powders exhibit stretching and bending vibration peaks of the O-H group at the 3400–3300 cm^−1^ and 1630–1620 cm^−1^ bands, respectively, with the peaks for CWO1 being particularly prominent. This observation indicates that a large number of hydroxyl groups are present on the surface of the CWO1 particles, suggesting a relatively hydrophilic character. It also implies that a small amount of water may be present in CWO1 during testing, which could result in other peaks being less distinct. In contrast, the hydroxyl peaks in powder 2 are relatively weak, indicating a more hydrophobic nature. Both powders exhibit strong absorption peaks for W-O-W and O-W-O bonds in the 1000–400 cm^−1^ range [34]. The redshift of these peaks is attributed to the distortion of the [WO6] octahedral lattice due to alkali metal doping [35,36]. Additionally, since Cs is fully incorporated into the WO_3_ lattice, no other peaks associated with Cs are observed throughout the spectrum.

#### 3.1.4. SEM

Figure 6 shows the SEM images of the two types of CWO powders and two-dimensional mapping based on Raman spectroscopy. The morphology of CWO1 is shown in Figure 6a, where the particles exhibit a hexagonal prismatic shape, with primary particle sizes ranging from approximately 50 to 500 nm, most commonly around 100 nm in height. This observation is consistent with the results calculated from the XRD tests, indicating a relatively large grain size, but with minimal severe agglomeration. The morphology of CWO2 is portrayed in Figure 6b, where the particles are significantly smaller, with primary particle sizes of about 10 to 50 nm. Although the primary particle size is smaller, the degree of agglomeration between the particles is much more pronounced. This is attributed to the fact that smaller CWO particles possess a relatively larger specific surface area, which results in increased surface energy. As a result, the attractive forces between atoms are stronger, promoting a higher likelihood of agglomeration, with the size of the secondary particles after agglomeration reaching approximately 1000 nm. SEM image analysis reveals that the morphology, grain size, and agglomeration states of the two types of CWO powders are different. To disperse them effectively, it is necessary to both disintegrate the larger grains and counteract the surface adsorption forces between particles to prevent agglomeration. The differences between the two powders determine that there will be significant variations in the challenges associated with their dispersion [5].

#### 3.1.5. TG-DSC

Figure 7 shows the TG-DSC curves of the two types of CWO powders. It is evident from the figure that neither of the CWO powders exhibits significant weight loss as the temperature increases, indicating that there are almost no organic residues present. The DSC curve indicates an endothermic process as the temperature rises, which can be attributed to the gradual decomposition of hexagonal tungsten bronze crystals at elevated temperatures [37]. Therefore, the application of CWO is subject to certain temperature limitations, as it is typically not functional at temperatures exceeding 300 °C.

#### 3.1.6. Size Distribution and Water Dispersion Spectra

Figure 8 illustrates the particle size distribution of the two types of CWO powders, as well as the particle size distribution of the dispersions obtained by ball milling the two powders in water under identical preparation conditions. As shown in the figure, the particle size distribution curves of CWO1 and CWO2 are consistent with the characterization analyses from XRD and SEM. CWO1 exhibits a larger average primary particle size but demonstrates less agglomeration, with a D50 value of 295 nm. Despite CWO2 having a smaller primary particle size, its significant agglomeration indicates a self-aggregation effect due to higher surface energy, resulting in a D50 of 1483 nm for the secondary particles. After a simple ball-milling process to form a dispersion, the D50 of powder 1 is 220 nm, whereas the D50 of powder 2, using the same process, is a higher 255 nm. These results indicate that although the crystals of powder 2 are finer, the particle size of the dispersion is even larger than that of powder 1 under the same process. This suggests that particle size is not the decisive factor for dispersibility, as factors like particle agglomeration and surface properties greatly influence the dispersion process. Therefore, it is crucial to select an appropriate dispersion process and system for different powders.

### 3.2. Influence of Dispersion Process

#### 3.2.1. Ball Size

Figure 9 shows the optical properties of dispersions prepared under different milling ball size configurations. As depicted in Figure 9a,c, the average particle size of the CWO1 dispersion decreases significantly with the reduction in milling ball size, leading to enhanced transmittance. Figure 9b,d show that the particle size of the CWO2 dispersion exhibits a similar trend with decreasing milling ball size. However, the average particle size of CWO2 is slightly larger, with a more concentrated particle size distribution. Concurrently, the visible light transmittance in its transmittance spectrum is extremely low. Based on the previous XRD and SEM analyses of the two powders, the reason for this phenomenon is that the primary particle size of CWO1 is larger compared to CWO2, with D50 = 458 nm, and it has a prismatic shape, allowing the abrasion to effectively break the grains and reduce the particle size. The resulting crushed particles are of an optimal size, and due to their compatibility with the aqueous medium, they exhibit a low tendency for agglomeration. In contrast, the primary particle size of CWO2 is already small, and the milling balls are unable to further break the grains, only serving to break up the agglomerates. However, without surface modification, the previously agglomerated particles tend to re-agglomerate after dispersion. Therefore, relying solely on physical dispersion methods is difficult to improve dispersibility significantly. Thus, for the water-based dispersion of CWO powder 1, using 0.1 mm milling balls can achieve good dispersion. However, for the ethanol-based dispersion of CWO powder 2, the milling ball size does not significantly improve the dispersion performance.

#### 3.2.2. Milling Time and Solid Content

Three solid content levels of 4%, 7%, and 11% were selected, and Figure 10 shows the performance of the dispersions after ball milling for 3 h, 5 h, and 7 h at a speed of 600 rpm. From Figure 10, it can be concluded that when the solid content is low, increasing the ball milling time does not have a significant effect on the optical performance of the dispersion, and in some cases, the performance may even decrease. This is because after the dispersion has been ball-milled to a certain extent, the shear force provided by the milling balls can no longer further break down the particles, and the tendency for agglomeration between particles increases, leading to a dynamic balance between particle fragmentation and agglomeration. Additionally, since mechanical ball milling generates some heat, excessive milling may oxidize the low-valence W in the CWO, resulting in performance degradation. Therefore, at lower solid content levels, considering the cost of time, equipment, and ball wear, a ball milling time of 3 h is optimal.

However, when the solid content is 11%, excellent optical performance can be achieved after 5 h of ball milling. Further increasing the milling time to 7 h does not result in significant performance changes, so a ball milling time of 5 h is ideal in this case.

After achieving optical properties in line with the targeted performance for the three solid content dispersions within the appropriate ball milling time, the next step is to compare their stability. Figure 11 shows the sedimentation diagrams of the three solid content dispersions after reaching their optimal optical performance through ball milling. After standing for 7 days, the 4% solid content dispersion showed no obvious stratification and had good stability. The 7% solid content sample appeared more turbid, with visible precipitation, and the liquid color was uneven between the upper and lower layers. The 11% solid content sample almost completely settled. This is because the higher the solid content of the dispersion, the more frequent the particle interactions, leading to greater agglomeration and consequently reducing the stability of the dispersion.

### 3.3. Influence of Dispersant

In addition to physical dispersion, the addition of dispersants is also an important method to improve dispersion effectiveness. The working mechanisms of dispersants can be divided into two types: first, the electrostatic stabilization mechanism, which enhances the electrostatic repulsion between particles by altering the surface charge properties of the particles; second, the steric stabilization mechanism, where dispersants are grafted onto the particle surface, utilizing steric hindrance to prevent particle-particle contact.

#### 3.3.1. Dispersant Type

Figure 12 and Figure 13 show the performance variation of dispersions after applying two different dispersants: Dispersant 1 and dispersant 4, which represent two different stabilization mechanisms. Dispersion system 1 refers to the CWO1/water/dispersant 1 system, while dispersion system 2 refers to the CWO2/ethanol/dispersant 4 system. For the two dispersion systems, several different kinds of dispersants were used, and their spectra are given in the figure. In system 1, the performance of the dispersion is optimal when dispersant 1 is used. The main component of dispersant 1 is sodium polyacrylate, a polyelectrolyte anionic dispersant with a molecular weight ranging from 2000 to 5000. It provides both electrostatic and steric stabilization mechanisms. Since the surface of CWO1 has a large number of graftable groups, the appropriate addition of this dispersant can effectively improve its dispersibility. Moreover, sodium polyacrylate has a high negative charge density due to its unique molecular structure. Compared to other dispersants of the same type, such as dispersant 2, whose main component is also polyacrylate, sodium polyacrylate is more effective at improving the potential of the double electron layer. In system 2, the addition of dispersant 4, which has PVB as its main component and a molecular weight of about 40,000, results in the best dispersion performance. This indicates that the grafting modification effect on CWO2 is not pronounced, but the steric dispersion effect of PVB is more effective. This is likely because there are fewer graftable groups on the surface of CWO2.

#### 3.3.2. Dispersant Dosage

Figure 14 shows the change in dispersion performance with different amounts of dispersant in two dispersion systems. From Figure 14a,c, it is evident that when a small amount (20%) of dispersant 1 is added, the optical properties of system 1 is improved. When the addition amount is too high, the adsorption capacity on the surface of CWO reaches saturation. At this point, free dispersant molecules connect with the nanoparticles to condense, which is detrimental to dispersion. Consequently, the optical performance is attenuated, and the stability is seriously reduced.

From Figure 14b,d, we can find a similar change. With the increase in dispersant 4 in system 2, the optical performance first improves and then decays, and the optimal amount is about 50%, but the mechanism causing this phenomenon is different. The main component of dispersant 4 is PVB, which is mainly a steric hindrance effect. When PVB is added appropriately, the resistance of particles in the dispersion to contact and reunite with each other will increase. However, due to the large molecular weight, viscosity of the liquid increases significantly when the addition reaches a certain amount, which makes the shear force provided by the grinding ball buffered during the dispersion process. Therefore, the performance of the dispersion is poor, especially the optical performance. Nevertheless, the high viscosity does not significantly reduce the stability of the dispersion, so the stability will maintain a good level with the increase in PVB content.

Through the above research, we optimized the CWO1/water (system 1) and CWO2/ethanol (system 2) to achieve excellent dispersion and optical selectivity, respectively. In system 1, the dispersion can maintain 87% T_550nm_ and 18% T_1100nm_; in system 2, 73% of T_550nm_ and 8% of T_1100nm_ can be maintained. Some of the differences are mainly related to the particle size distribution caused by various factors.

### 3.4. Simulation of CWO Dispersion

In the aforementioned study, the dispersion liquids prepared from two types of powder achieved good dispersion effects through optimization from multiple angles. However, during the tests, it was observed that after reaching a certain degree of optimization, the spectral performance of the two dispersions became largely similar and sufficiently stable, with minor differences in the 550–950 nm wavelength range. This phenomenon may primarily be related to the difference in elemental valence states between the two powders. CWO1 contains more W^5+^ ions, leading to a certain level of oxygen vacancy, which enhances its small polaron absorption effect. Nonetheless, this phenomenon may not have a significant impact on the overall spectrum of the dispersion. In addition, the differences in dispersing medium and particle size distribution might be the main factors contributing to this phenomenon. Therefore, based on the LSPR absorption model, COMSOL 6.1 software was used to simulate the optical absorption performance of the nano CWO particles in the two dispersions for verification. A quarter model was established based on the symmetry of the particles, as shown in Figure 15. Figure 15a represents CWO1 particles, which are hexagonal prisms. Due to uniform ball milling, the aspect ratio can be approximated as 1. Moreover, the particles have high crystallinity, and various parameters of the LSPR model exhibit anisotropy. Figure 15b represents CWO2 particles, which are aggregates of several small spherical particles and can thus be approximated as homogeneous isotropic spheres.

In the medium domain and powder particle domain, the wave equations are defined by Equations (3) and (4), respectively, where **E** is the electric field intensity, $E=e^{-2\pi jx/\lambda }$, $\varepsilon_{r}$ is the refractive index, *j* is the imaginary unit, λ is the wavelength, and $\mu_{r}$ is the relative magnetic permeability.
(3)$$\nabla \times (\nabla \times E)-k_{0}^{2}\varepsilon_{r}E=0,$$
(4)$$\nabla \times \mu_{r}^{-1}(\nabla \times E)-k_{0}^{2}\varepsilon_{r}E=0,$$

The scattering boundary condition of the medium domain is defined by Equation (5), and the symmetric boundary condition of the powder particle cross section is defined by Equation (6), where **n** is the normal vector, and k is the imaginary unit.
(5)n×(∇×E)−jkn×(E×n)=0,
(6)n×E=0

Figure 16, Figure 17, Figure 18 and Figure 19 present the simulated results of the absorption coefficient mentioned above, with the calculation wavelength range set to 300–1100 nm. Calculations were performed at intervals of 50 nm, and the results for different wavelengths at the same particle size were plotted as scatter points in Origin 2018. The modified Bezier method was used to connect and fit the scatter points into curves.

The simulation results of CWO particles in water and ethanol at four different particle size scales are shown in Figure 16. It can be observed that within the representative particle size scales under discussion, there is almost no difference in the light absorption spectra of the particles in the two dispersing media below the 950 nm wavelength. In the 950–1100 nm wavelength range, there is a relatively weak effect. Therefore, the spectral differences in the dispersion medium in the 650–900 nm wavelength range, which is the focus of this study, are largely unrelated to the type of medium.

As shown in Figure 17 and Figure 18, both CWO powders 1 and 2 exhibit significant changes in absorption coefficients with particle size. Additionally, at the same particle size, the absorption coefficient curves of CWO1 and CWO2 show noticeable differences. This is attributed to the anisotropy of the particle shape and material parameters. Figure 18 presents the average normalized absorption coefficient obtained by performing weighted calculations on the curves from Figure 17 and Figure 18. The weighting factor used is the particle size distribution measurement of the dispersion liquid, i.e., the frequency at which each particle size appears in the dispersion. This weighted averaging method allows a simple approximation of the light absorption performance of the dispersion liquid. The results indicate that the absorption capacity of the CWO particles in dispersion system 1 is slightly stronger than that of system 2 in the 550–950 nm wavelength range. This difference in absorption capacity is due to the differences in particle size and shape shown in Figure 17 and Figure 18 and is consistent with the measured data. The dashed lines represent the measured absorption spectra of the two dispersion liquids, where similar differences can be observed. Thus, this validates that the subtle differences between the two dispersion liquids in this wavelength range are related to the particle shape and the differences in particle size distribution caused by the preparation process.

Additionally, although the experimental results are validated within the scope of this study when compared with the simulation results, the spectra do not fully align. The simulation results show a noticeable upward shift and even exhibit almost no correlation in the lower wavelength range. This discrepancy arises because the model is aimed at a qualitative analysis of the absorption performance differences in the dispersions, without accounting for the impact of light scattering. Given that the particle size distribution of both dispersions ranges from 50 to 1000 nm, light with a wavelength below 1000 nm is significantly affected by scattering, which reduces the transmittance. The shorter the wavelength, the more complex the scattering becomes, and the more pronounced the scattering loss is. As a result, the deviation from the experimental data increases.

## 4. Conclusions

In this paper, a series of CWO dispersions were systematically prepared and analyzed. The comprehensive effects of powder type, ball milling process, and dispersant on the dispersion and optical properties of CWO were discussed. Through experimental tests and preliminary simulation verification, the main conclusions of this study are as follows:CWO1, obtained by the modified solid-phase method, is more suitable for aqueous dispersion, while CWO2 is suitable for ethanol with weak polarity.For CWO1, which has coarse primary particles and light agglomeration, the size of the grinding ball significantly affects the dispersion effect. Using a 0.1 mm grinding ball to disperse for 3 h can greatly improve the performance of the dispersion with a 4% solid content, achieving an average particle size of 105 nm. In contrast, for CWO2, which has fine primary particles but a serious agglomeration tendency, the effect of mechanical ball milling on particle dispersion is not significant enough, and the average particle size can only reach 190 nm.When combined with sodium polyacrylate at 20% of the powder mass, CWO1 showed the best performance, maintaining 87% transmittance at 550 nm (T_550nm_) and 18% transmittance at 1100 nm (T_1100nm_). Similarly, the effect of CWO2 combined with 50% PVB powder mass was significantly improved, maintaining 73% T _550nm_ and 8% T_1100nm_.During the experiment, it was found that although both CWO1 and CWO2 dispersions showed excellent visible light transmittance and near-infrared shielding properties, the dispersion prepared with CWO1 had higher shielding properties in the 650–950 nm range. Analysis revealed that these subtle differences are mainly due to the difference of CWO shape and size distribution after dispersion, which is supported by simulation calculation based on the Drude–Lorenz model and the finite element method (FEM).

## 5. Future Works

Cs_x_WO_3_ is a water-based functional slurry with high visible light transmittance and high infrared shielding. The water-based slurry can be applied to architectural glass through processes such as casting, coating, and spraying. Architectural glass containing CsxWO3 slurry has good lighting. At the same time, due to the high infrared shielding of CsxWO3 slurry architectural glass, the indoor temperature of the building maintains a certain stability, reducing the energy consumption of the building. At present, there is little comprehensive application research on tungsten bronze coatings, films, and other composite materials, and there are still gaps in some areas, for example, the performance research of tungsten bronze laminated glass; whether and how tungsten bronze materials can fully exert their excellent performance in multi-component coating and film material systems; and how new tungsten bronze energy-saving glass can be used in conjunction with traditional energy-saving glass, etc. These issues deserve further research.

## Figures and Tables

**Figure 1 materials-17-05196-f001:**
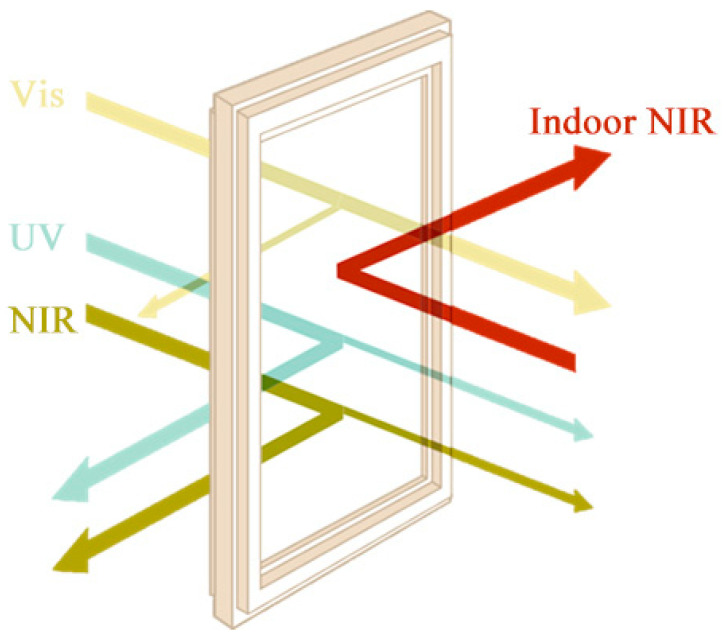
The work principle of energy-saving glass using transparent insulation materials [5].

**Figure 2 materials-17-05196-f002:**
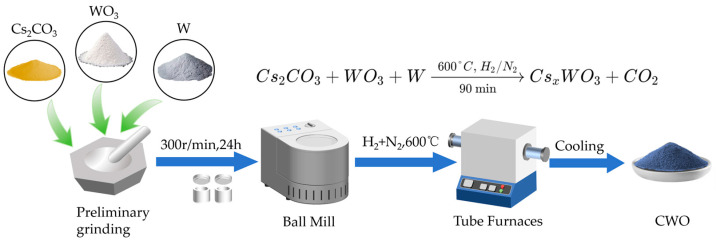
Schematic diagram of CWO powder preparation.

**Figure 3 materials-17-05196-f003:**
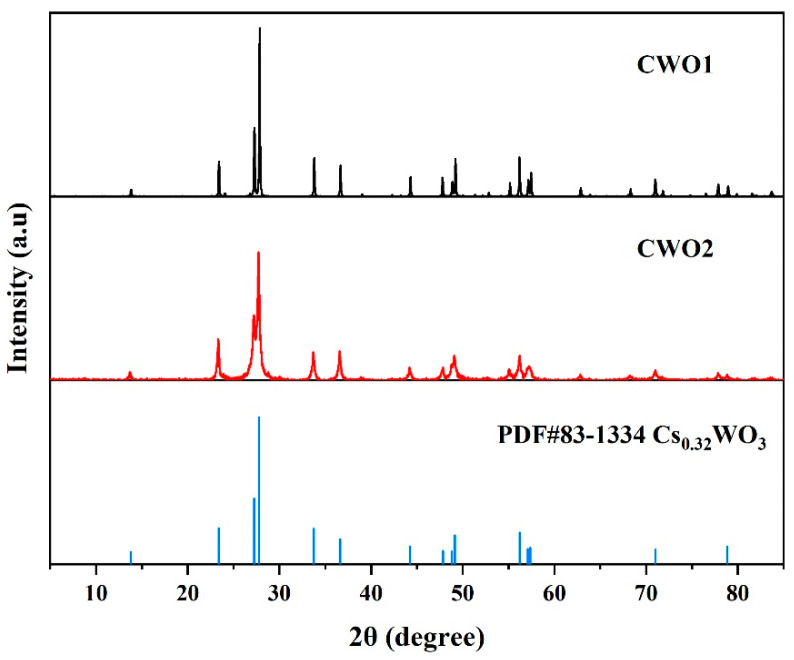
XRD of CWO powder.

**Figure 4 materials-17-05196-f004:**
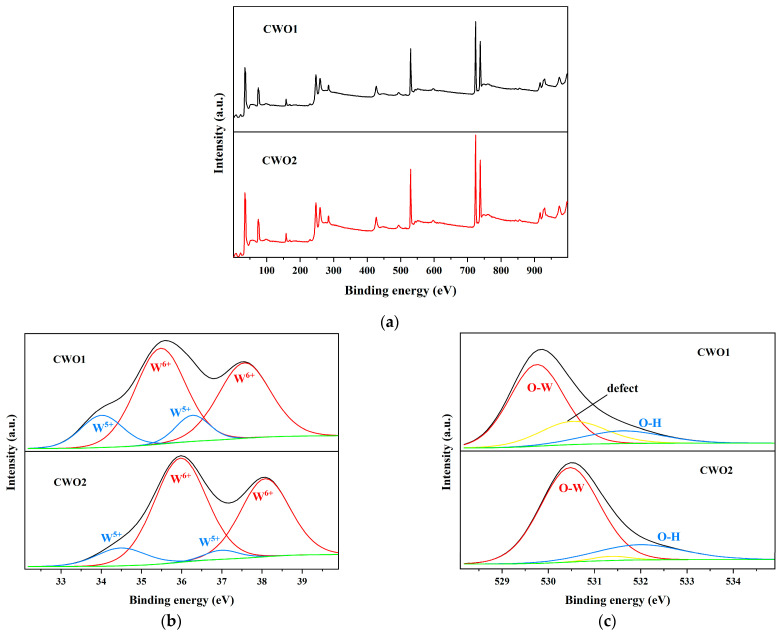
XPS of CWO powder: (**a**) Survey spectra; (**b**) W4f spectra; (**c**) O1s spectra.

**Figure 5 materials-17-05196-f005:**
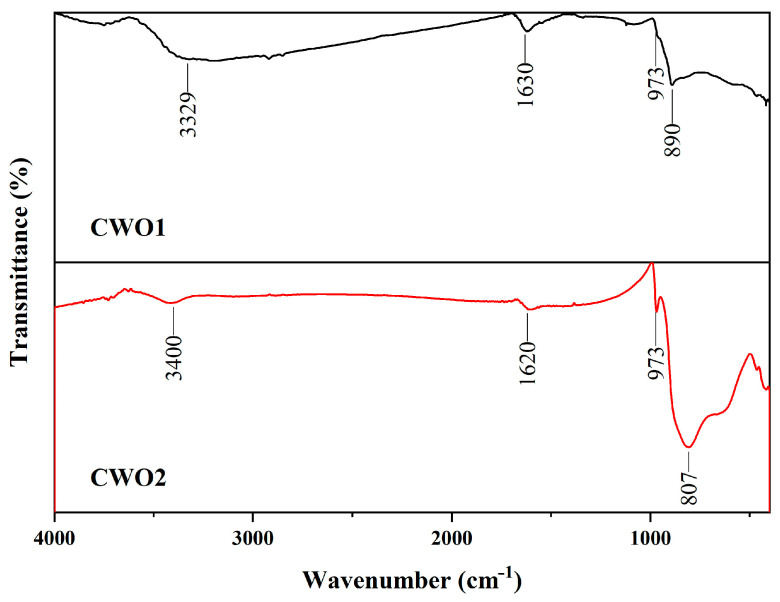
FT-IR spectra of CWO powder.

**Figure 6 materials-17-05196-f006:**
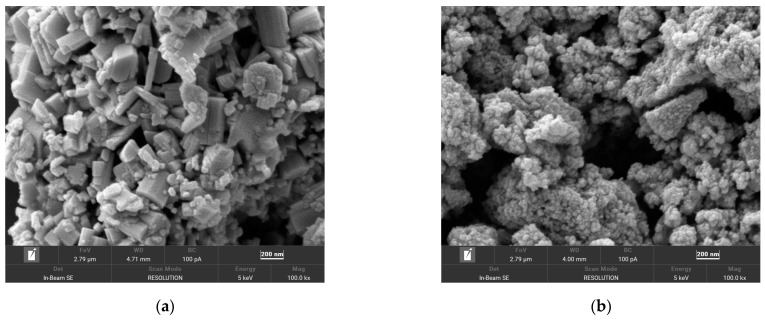
SEM image of CWO powder: (**a**) CWO1; (**b**) CWO2.

**Figure 7 materials-17-05196-f007:**
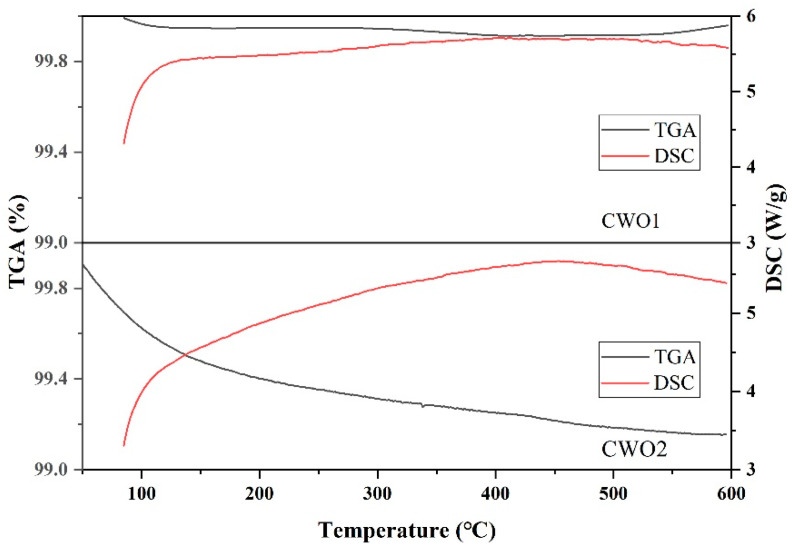
TG-DSC of CWO powder.

**Figure 8 materials-17-05196-f008:**
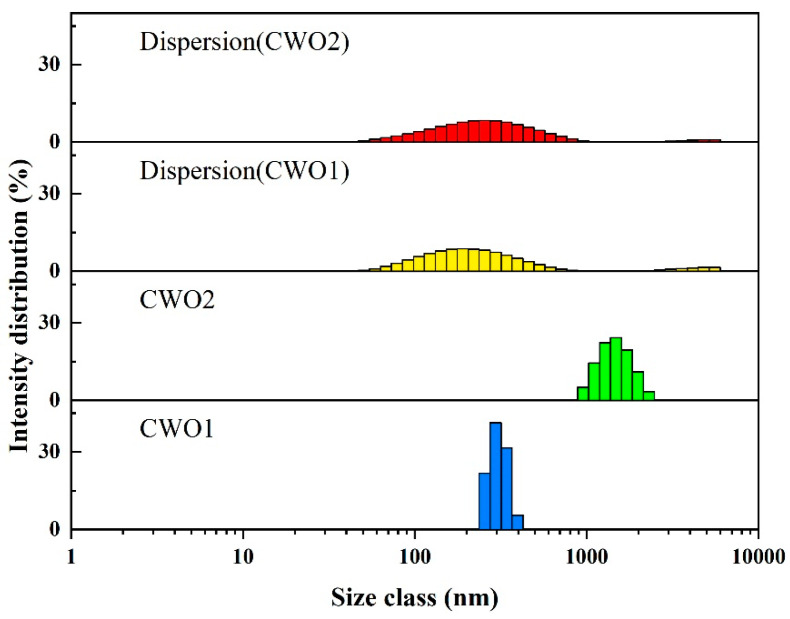
Particle size of powder and aqueous dispersion.

**Figure 9 materials-17-05196-f009:**
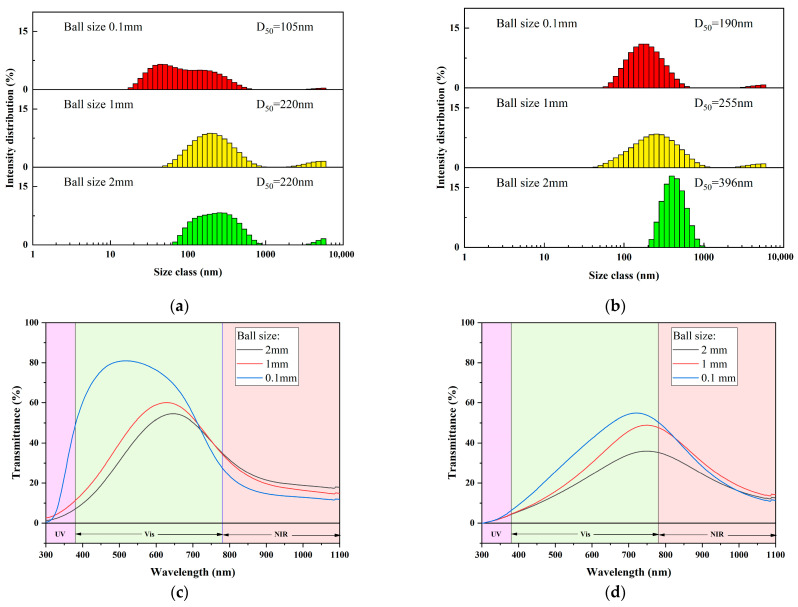
Properties of dispersion with different ball sizes: (**a**) Particle size of CWO1 dispersion; (**b**) Particle size of CWO2 dispersion; (**c**) Transmittance spectra of CWO1 dispersion; (**d**) Transmittance spectra of CWO2 dispersion.

**Figure 10 materials-17-05196-f010:**
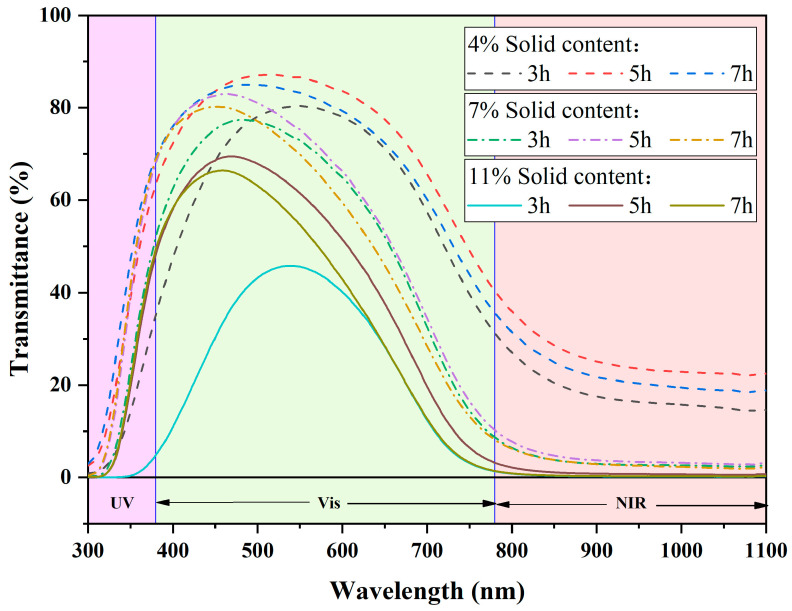
Transmission spectra of dispersions with different solid contents and grinding times.

**Figure 11 materials-17-05196-f011:**
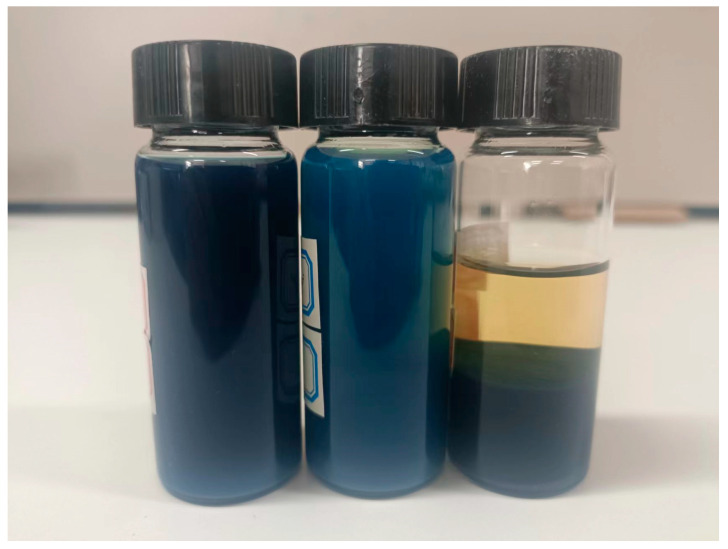
Stability of dispersions with different solid contents and grinding times.

**Figure 12 materials-17-05196-f012:**
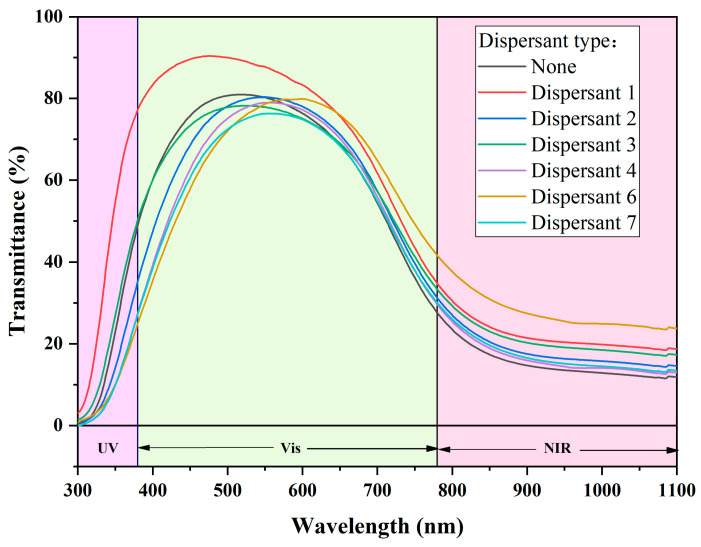
Dispersion spectrum of dispersion system 1.

**Figure 13 materials-17-05196-f013:**
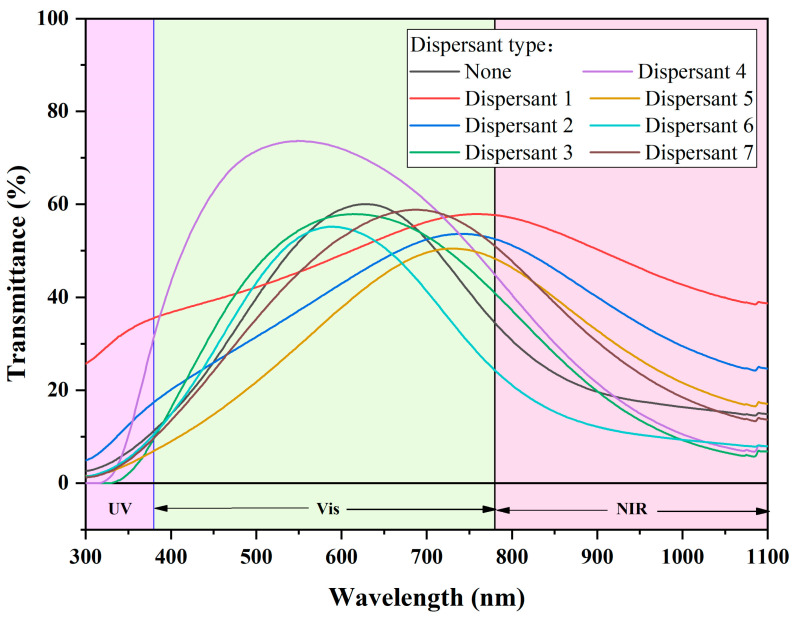
Dispersion spectrum of dispersion system 2.

**Figure 14 materials-17-05196-f014:**
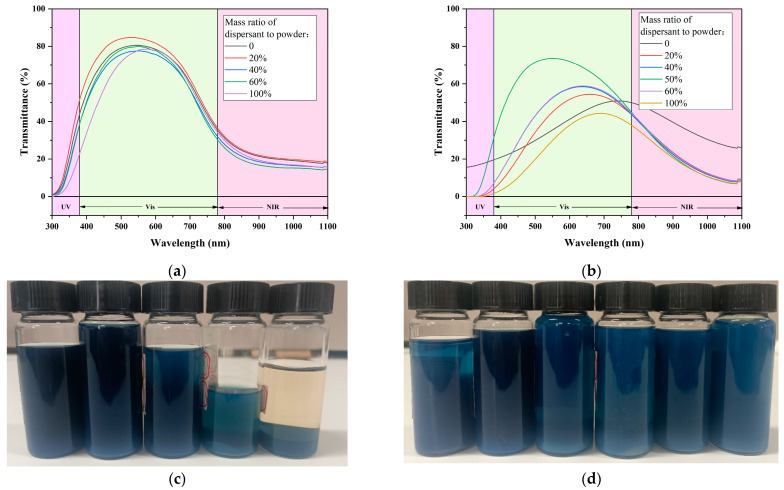
Properties with different dispersant dosage: (**a**) Dispersion system 1—spectra; (**b**) Dispersion system 2—spectra; (**c**) Dispersion system 1—stability; (**d**) Dispersion system 2—stability.

**Figure 15 materials-17-05196-f015:**
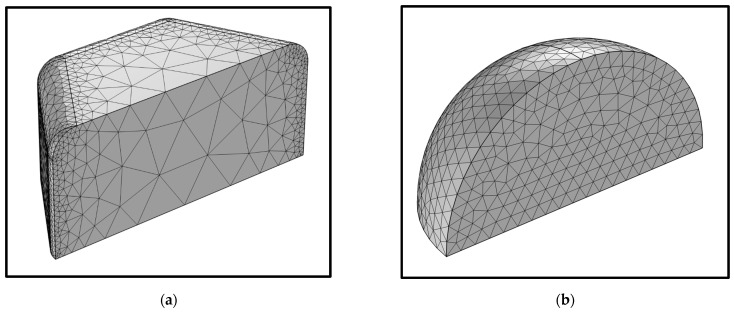
Mesh of model of nano CWO particle in dispersion: (**a**) Dispersion system 1; (**b**) Dispersion system 2.

**Figure 16 materials-17-05196-f016:**
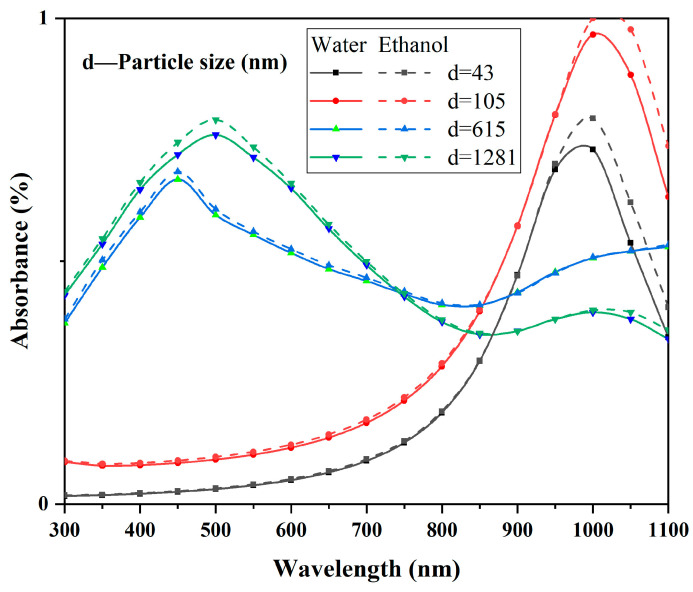
Effect of medium on LSPR absorption of CWO nanoparticles.

**Figure 17 materials-17-05196-f017:**
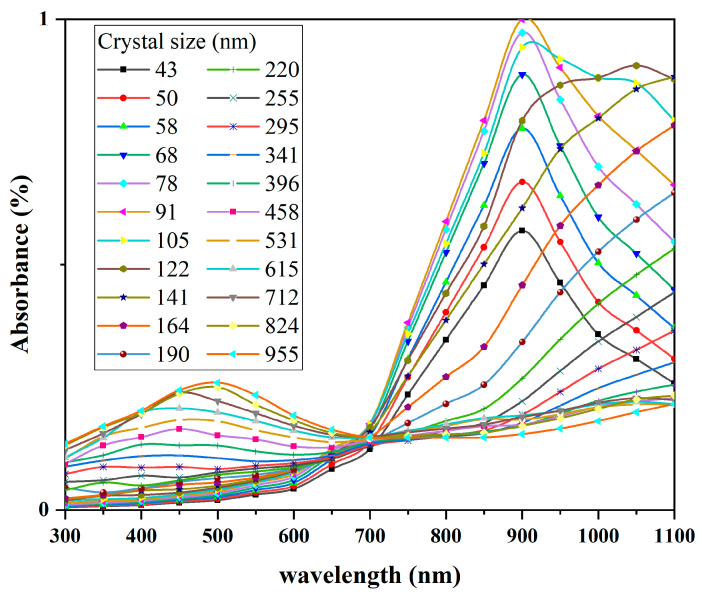
LSPR absorption simulation results of CWO1 with different particle sizes.

**Figure 18 materials-17-05196-f018:**
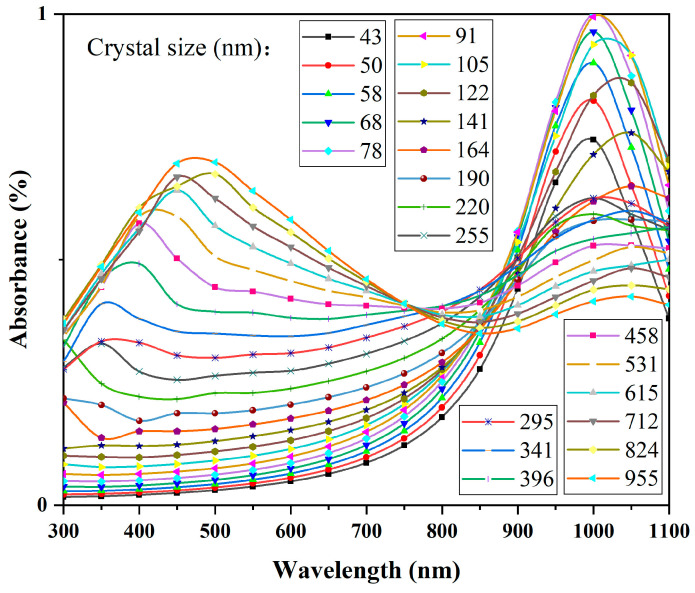
LSPR absorption simulation results of CWO2 with different particle sizes.

**Figure 19 materials-17-05196-f019:**
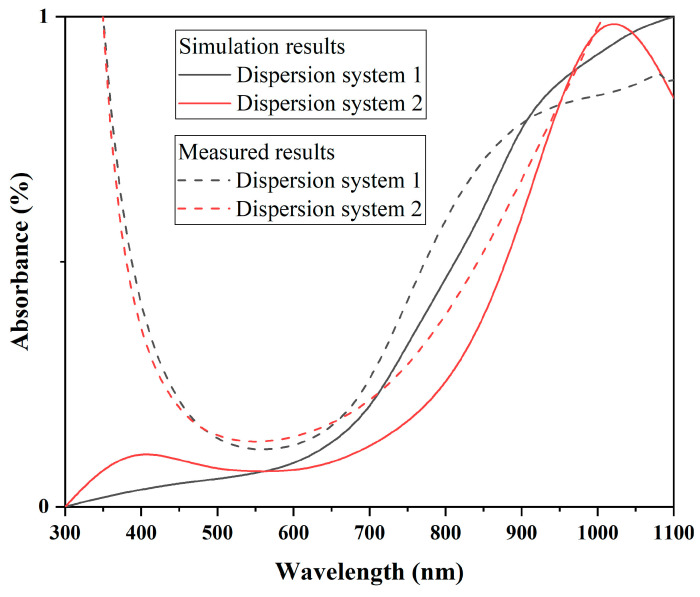
Weighted average absorption curve using particle size distribution.

**Table 1 materials-17-05196-t001:** Dispersant information.

Name	Pureness	Source	Note
A4100	Tech	Jinan Puluosi New Materials Co., Ltd. (Jinan, China)	Dispersant 1
TEGO-735w	Tech	Shenzhen Longdi Chemical Co., Ltd. (Shenzhen, China)	Dispersant 2
PVP	AR	Aladdin Scientific Corp. (Riverside, CA, USA)	Dispersant 3
PVB	AR	Weng Jiang Reagent Co., Ltd. (Weng Jiang, China)	Dispersant 4
PVA	AR	Rhawn Reagent (Shanghai, China)	Dispersant 5
Sodium citrate	AR	Xiya Reagent (Jinan, China)	Dispersant 6
Win4196	Tech	Shandong Winbos New Materials Co., Ltd. (Weifang, China)	Dispersant 7

**Table 2 materials-17-05196-t002:** Material parameters of CWO particles.

Material Parameters	Longitudinal	Transverse
High frequency dielectric constant $\varepsilon_{\infty }$	6.3	5.8
Plasma frequency $\omega_{p}$ (rad/s)	7.0844 × 10^15^	4.8316 × 10^15^
Scattering constant γ (rad/s)	3.2964 × 10^14^	5.0859 × 10^14^

## Data Availability

The data presented in this study are available on request from the corresponding author (due to privacy).

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
