# Peer review of "Properties and Factors of CsxWO3 Slurry for Building Glass with High Visible Light Transmission and Outstanding Near-Infrared Insulation"

_materials, 2024, doi:10.3390/ma17215196_

Round 1

Reviewer 1 Report

Comments and Suggestions for Authors

Referee report on manuscript “Properties and factors of CsxWO3 slurry for building glass  with high visible light transmission and excellent near-infare insulation”

This is an interesting article, it can be certainly recommended for publication after clarifying some uncertainties.

1.     Line 44. The end of this sentence needs supporting references.

2.     Line 67. The end of this sentence needs few supporting references.

3.     Line 86.  For sentence “The plasma vibrates at a certain natural frequency in crystal“, more detailed explanation is required.

4.     Line 105. The end of this sentence needs supporting references. Moreover, the mentioned different preparation methods lead to different levels of point defects, which will be reflected in the optical properties. An appropriate clarification is required.

5.     Based on the above, a few lines about the current status of understanding point defects would be useful to give to the readers.

6.     Paragraph 3.1.4. SEM.  This part of the article suggests the need to conduct 2-dimensional mapping, for example, based on Raman spectroscopy.

7.     How important and necessary is it to conduct nanoporosity analysis for these glasses?

8.     It will nice to see data on optical and thermal stability (aging effects), which is important for practical applications.

9.     In the conclusions section, it would be absolutely necessary to formulate clearly what fundamentally new data were obtained for the material studied.

Author Response

Comments 1: 1.     Line 44. The end of this sentence needs supporting references.

Response 1: Thank you for pointing this out. We agree with this comment.We added three recent references to support this sentence.

Comments 2: 2.     Line 67. The end of this sentence needs few supporting references.

Response 2: Thank you for pointing this out. We agree with this comment.We added four recent references to support this sentence.

Comments 3: 3.     Line 86.  For sentence “The plasma vibrates at a certain natural frequency in crystal“, more detailed explanation is required.

Response 3:Thank you for pointing this out. We agree with this comment.In line 98, we explain this sentence in detail.

Comments 4: 4.     Line 105. The end of this sentence needs supporting references. Moreover, the mentioned different preparation methods lead to different levels of point defects, which will be reflected in the optical properties. An appropriate clarification is required.

Response 4: Thank you for pointing this out. We agree with this comment.

This sentence comes from the summary of the above literature review, so no references are added at the end to support it.

Comments 5: 5.     Based on the above, a few lines about the current status of understanding point defects would be useful to give to the readers.

Response 5:

Thank you for pointing this out. We agree with this comment.Maybe I didn't express it correctly. This paragraph mainly talks about that LSPR absorption effect is related to the shape, size and medium of tungsten bronze particles. Subsequent experiments and simulations are also based on this conclusion, and the concept of point defects is not involved.

Comments 6: 6.     Paragraph 3.1.4. SEM.  This part of the article suggests the need to conduct 2-dimensional mapping, for example, based on Raman spectroscopy.

Response 6:

Thank you for pointing this out. We agree with this comment.We have modified it in the article.

Comments 7: 7.     How important and necessary is it to conduct nanoporosity analysis for these glasses?

Response 7: Thank you for pointing this out. We agree with this comment.Your suggestion is very valuable for me. We will discuss this direction in detail in the next step of our work.

Comments 8: 8.     It will nice to see data on optical and thermal stability (aging effects), which is important for practical applications.

Response 8: Thank you for pointing this out. We agree with this comment.Your suggestion is very valuable for me. We will discuss this direction in detail in the next step of our work.

Comments ï¼™: 9.     In the conclusions section, it would be absolutely necessary to formulate clearly what fundamentally new data were obtained for the material studied.

Response 9:

Thank you for pointing this out. We agree with this comment.At the end, we give the powder addition amount for better dispersion effect, as well as the visible light transmittance and near-infrared shielding under this ratio, and support it through simulation calculations based on the Drude-Lorenz model and finite element method.

Reviewer 2 Report

Comments and Suggestions for Authors

The abstract is OK drafted

The introduction is well written

All the acronyms should be defined before their first apparency in the text i.e. SEM and many other

The boundary conditions of model used in 2.2.4 should be clearly indicate as well as all other loading and mesh process.

The text should be checked for different typos “in CWO1, This suggests “

“of CWO1 is relatively high” in respect to what – as well as a quantitative description is required

“can achieve ideal optical properties 291 after appropriate ball milling time2 not sure if ideal properties can be claimed rather than may be it will be worth indicating that the condition achieved is lined with the targeted properties.

The conclusion are OK

Some recent literature is required as it can demonstrate the novelty of the work, as now many of the literature stated are very old..

Comments on the Quality of English Language

some improvements is required

Author Response

Comments 1: 1.     All the acronyms should be defined before their first apparency in the text i.e. SEM and many other

Response 1: Thank you for pointing this out. We agree with this comment.All acronyms are defined, such as SEM, XRD, XPS, etc.

Comments 2: 2.    The boundary conditions of model used in 2.2.4 should be clearly indicate as well as all other loading and mesh process.

Response 2: Thank you for pointing this out. We agree with this comment.We have modified it in the article.

Comments 3: 3.    The text should be checked for different typos “in CWO1, This suggests “

“of CWO1 is relatively high” in respect to what – as well as a quantitative description is required

“can achieve ideal optical properties 291 after appropriate ball milling time2 not sure if ideal properties can be claimed rather than may be it will be worth indicating that the condition achieved is lined with the targeted properties.

Response 3:Thank you for pointing this out. We agree with this comment.We have made targeted modifications to the corresponding locations in the article

Comments 4: 4.     

Some recent literature is required as it can demonstrate the novelty of the work, as now many of the literature stated are very old..

Response 4: Thank you for pointing this out. We agree with this comment.We have added nearly ten new references to the Introduction section

Reviewer 3 Report

Comments and Suggestions for Authors

I believe that after minor corrections, the work will be suitable for publication.

The Manuscript topic says, "Properties and factors of CsxWO3 slurry for building glass..." However, in the body of the paper, there are no references to "building glass." Also, the Conclusions section does not contain information on this topic. I ask the authors to supplement the body of the Manuscript with this information and discuss the possibilities of applying their work in practice.

The Introduction chapter could include more newer literature. Please consider this.

Line 128 - CWO preparation - a diagram showing the next production steps would be helpful here.

Lines 141 – 156  Do these devices have any accuracy class? Please provide.

Figure 9(a) is poorly visible; I think it is too small. Please correct it.

Figures 10 and 13 are poorly visible and too small. Arranging the drawings one below the other might help. Please correct it.

References:

Please provide DOI numbers where it is possible.

Author Response

Comments 1: 1.     The Manuscript topic says, "Properties and factors of CsxWO3 slurry for building glass..." However, in the body of the paper, there are no references to "building glass." Also, the Conclusions section does not contain information on this topic. I ask the authors to supplement the body of the Manuscript with this information and discuss the possibilities of applying their work in practice.

Response 1: We discussed the effects of CsxWO3 slurries as coatings on architectural glass in a previous publication and for the sake of completeness, we cite this paper to illustrate the feasibility of their work in practice.

Comments 2: 2.    The Introduction chapter could include more newer literature. Please consider this.

Response 2: Thank you for pointing this out. We agree with this comment.We have added nearly ten new references to the Introduction section

Comments 3: 3.    Line 128 - CWO preparation - a diagram showing the next production steps would be helpful here.

Response 3:Thank you for pointing this out. We agree with this comment.We updated Figure 2 to describe the material production process

Comments 4: 4.     Lines 141 – 156  Do these devices have any accuracy class? Please provide.

Response 4: Thank you for pointing this out. We agree with this comment.In Section 2.2.2, we added the accuracy level of the device.

Comments 5: 5.     Figure 9(a) is poorly visible; I think it is too small. Please correct it.Figures 10 and 13 are poorly visible and too small. Arranging the drawings one below the other might help. Please correct it.

Response 5: Thank you for pointing this out. We agree with this comment.We have arranged these figures individually

Comments 6: 6.     Please provide DOI numbers where it is possible.

Response 6:Thank you for pointing this out. We agree with this comment.We have added the DOI of the article

Round 2

Reviewer 1 Report

Comments and Suggestions for Authors

After successful revision, this manuscript can be recommended for publication.

Author Response

Thank you very much for your valuable comments and support for our work!

Reviewer 2 Report

Comments and Suggestions for Authors

no further comments 

Comments on the Quality of English Language

.

Author Response

(The authors gave the same response as above.)
